# Antipodal Pairing and Mechanistic Signals in Dense SAE Latents

**Alessandro Stolfo**[1][*]  **Ben Wu**[2]  **Mrinmaya Sachan**[1]

[1]ETH Zürich  [2]University of Sheffield

## Abstract

Sparse autoencoders (SAEs) are designed to extract interpretable features from language models, yet they often yield frequently activating latents that remain difficult to interpret. It is still an open question whether these *dense* latents are an undesired training artifact or whether they represent fundamentally dense signals in the model's activations. Our study provides evidence for the latter explanation: dense latents capture fundamental signals which (1) align with principal directions of variance in the model's residual stream and (2) reconstruct a subspace of the unembedding matrix that was linked by previous work to internal model computation. Furthermore, we show that these latents typically emerge as nearly antipodal pairs that collaboratively reconstruct specific residual stream directions. These findings reveal a mechanistic role for dense latents in language model behavior and suggest avenues for refining SAE training strategies.

## 1 Introduction

Sparse autoencoders (SAEs) offer an unsupervised method for extracting interpretable features from language models (Bricken et al., 2023; Huben et al., 2024; Kissane et al., 2024). They address the challenge of polysemanticity, where individual neurons activate in semantically diverse contexts that defy simple explanation (Olah et al., 2017; Elhage et al., 2022). SAEs are trained to reconstruct the activations of a language model under a sparsity constraint applied to a bottleneck layer, ensuring that only a small subset of latents is active at a time.[1] This method was shown to effectively recover interpretable features in a variety of models, including Claude 3 Sonnet (Templeton et al., 2024) and GPT-4 (Gao et al., 2025).

Ideally, training would yield a large set of latents that activate sparsely and in an interpretable manner. In practice, however, most SAEs exhibit densely activating latents, with some activating on more than 10% and up to 50% of tokens (Cunningham & Conerly, 2024; Rajamanoharan et al., 2024). These dense activations are challenging to interpret based solely on their patterns, and it remains unclear whether they arise as an optimization by-product or if they capture inherently dense signals present in the model's residual stream.

In this work, we examine the relationship between the activation frequency of SAE latents and their composition with specific subspaces of the residual stream. We analyze of SAEs trained on Gemma 2 (Gemma Team, 2024), GPT-2 (Radford et al., 2019), and LLaMA 3.1 (AI @ Meta, 2024), finding that densely activating latents tend to read from and write to the space spanned by the top principal components of the residual stream. Further, we observe that some of these directions align with the bottom singular vectors of the model's unembedding matrix–a subspace previously linked to internal computation signals (Cancedda, 2024) such as confidence regulation (Stolfo et al., 2024). We also show that most dense latents are arranged in antipodal pairs, with encoder and decoder weights operating in nearly opposite directions, reconstructing specific directions within this subspace.

Our findings provide evidence that SAEs learn to allocate a subset of their latents to precisely reconstruct specific directions in the residual stream. These directions appear to represent signals that are fundamentally dense and play a mechanistic role rather than conveying semantic properties of

---

[*]Correspondence to stolfoa@ethz.ch.
[1]We use "latent" to refer to an entry in the SAE's sparse hidden layer.

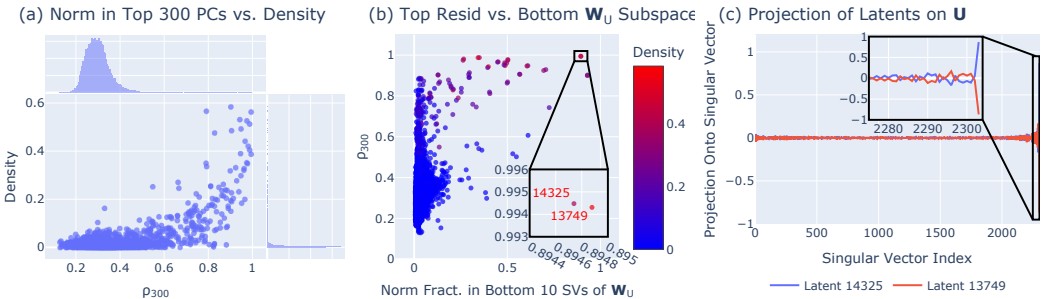

Figure 1: **Dense latents align with key residual subspaces.** (a) Activation density vs. norm fraction in top 300 residual PCs. (b) Norm composition in top residual PCs vs. bottom $\mathbf{W}_U$ singular vectors, highlighting paired outliers. (c) Projection of two dense latents onto $\mathbf{W}_U$ singular vectors.

the input. This work contributes to a better understanding of SAE training and lays the groundwork for future research on SAE methods to more effectively account for these dense signals.

## 2 BACKGROUND

Sparse autoencoders (SAEs) are trained to reconstruct a language model's activations $\mathbf{x} \in \mathbb{R}^{d_{\mathrm{model}}}$ while imposing a sparsity constraint (Yun et al., 2021; Huben et al., 2024). This computation can be represented as:

$$\mathbf{f}(\mathbf{x}) := \sigma(\mathbf{W}_{\mathrm{enc}}\mathbf{x} + \mathbf{b}_{\mathrm{enc}}),$$
$$\hat{\mathbf{x}}(\mathbf{f}) := \mathbf{W}_{\mathrm{dec}}\mathbf{f} + \mathbf{b}_{\mathrm{dec}},$$

where $\mathbf{f}(\mathbf{x}) \in \mathbb{R}^{d_{\mathrm{sae}}}$ is a sparse, non-negative vector of latents, with $d_{\mathrm{sae}} >> d_{\mathrm{model}}$, and $\sigma$ is a non-linear activation function. SAEs are typically trained to minimize the L2 distance between the original activation and its reconstruction $\|\mathbf{x} - \hat{\mathbf{x}}(\mathbf{f}(\mathbf{x}))\|_2^2$ while a sparsity constraint is imposed on $\mathbf{f}$ by adding a sparsity-related loss component or via specific activation functions. We denote the encoder and decoder weights of the latent at index $i$ as $\mathbf{W}_{\mathrm{enc}}^{(i)}$ and $\mathbf{W}_{\mathrm{dec}}^{(i)}$, respectively. Unless noted otherwise, we use "dense" to refer to latents with an activation frequency larger than 0.1.

We focus our investigation on the Gemma Scope SAEs (Lieberum et al., 2024) trained on Gemma 2 2B (Gemma Team, 2024), which use a JumpReLU activation function (Rajamanoharan et al., 2024). However, we provide results also for TopK SAEs (Gao et al., 2025) trained on the activations of GPT-2 (Radford et al., 2019) and LLaMA 3.1 8B (AI @ Meta, 2024).

## 3 EMPIRICAL ANALYSES

### 3.1 DENSE LATENTS ALIGN WITH DOMINANT RESIDUAL-STREAM SUBSPACE

Because SAEs are trained under a constraint that limits the number of active latents at any given time, the presence of densely activating latents represents a significant allocation of representational capacity. This observation implies that these latents encode signals that are important for minimizing reconstruction loss. We therefore hypothesize that such signals are concentrated along the residual stream directions that account for most of its variance, namely, the top principal components.

To test this hypothesis, we compute the top $k$ principal components of the Gemma 2 2B residual stream over approximately 250k tokens of the C4 Corpus (Raffel et al., 2020). We then quantify the "composition" of an SAE latent $i$ with the top $k$ subspace by computing the fraction $\rho_k$ of the norm of its encoder weight $\mathbf{W}_{\mathrm{enc}}^{(i)}$ that lies in this subspace:

$$\rho_k = \frac{\sum_{j=1}^{k} \mathbf{v}_j^{\mathrm{T}} \mathbf{W}_{\mathrm{enc}}^{(i)}}{\|\mathbf{W}_{\mathrm{enc}}^{(i)}\|}, \tag{1}$$

where $\mathbf{v}_i$ indicates the $i$-th principal component. In Figure 1a, we report this composition against the activation density for $k = 300$ for each latent in the 16k-latent SAE trained on the residual

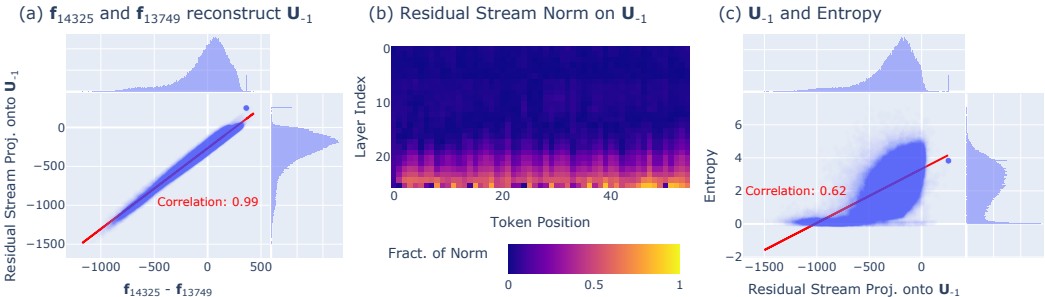

Figure 2: **Dense latents reconstruct a key residual direction.** (a) Latent pair activation difference vs. residual projection on the last $\mathbf{W}_U$ singular vector $\mathbf{U}_{-1}$. (b) Residual norm fraction in this direction across layers and tokens for a single sequence. (c) Correlation between residual stream value along $\mathbf{U}_{-1}$ and model entropy across 100k tokens.

stream at layer 25 (0-indexed) of Gemma 2 2B. The plot reveals a clear trend: 99% of latents exhibit a composition less than 0.7 and an activation density below 0.1, whereas outliers with high composition ($\rho_{300} > 0.7$)–of which 67% are densely activating–account for 82% of all densely activating latents.

## 3.2 DENSE LATENTS ALIGN WITH THE $\mathbf{W}_U$ DARK SUBSPACE

Prior work on language model interpretability has identified an interaction between high-norm model components and the *dark* subspace spanned by the last singular vectors of the unembedding matrix $\mathbf{W}_U$ (Stolfo et al., 2024; Cancedda, 2024). Given that this subspace absorbs a substantial portion of the residual stream's norm, we hypothesize that some of the top variance directions in the residual stream may coincide with the bottom $\mathbf{W}_U$ subspace.

To investigate this possibility, we compute the singular value decomposition $\mathbf{W}_U = \mathbf{U}\mathbf{\Sigma}\mathbf{V}^T$. Then, we study the composition of each latent's decoder weights with the top residual stream principal components against its composition with the bottom 10 left singular vectors $\mathbf{U}_{-10}, \ldots, \mathbf{U}_{-1}$ of $\mathbf{W}_U$ (Figure 1b).[2] The analysis shows that latents with high composition in the top residual subspace also tend to have high composition in the bottom $\mathbf{W}_U$ subspace.

Furthermore, the outliers–latents with high composition in both subspaces–appear to be clustered in pairs. On closer inspection, these pairs exhibit antipodal encoder and decoder weights; that is, the cosine similarity between the corresponding encoder vectors (and likewise for the decoder vectors) is close to $-1$. Figure 1c illustrates an example where the latent weights are highly aligned with the last singular vector $\mathbf{U}_{-1}$. Although these paired latents are densely activating, they are virtually never active simultaneously (Appendix B, Figure 5b). This observation suggests that the pair collectively reconstructs points along a specific line: the direction defined by the $\mathbf{U}_{-1}$. In Figure 2a, we plot the difference in activation values $\mathbf{f}_i - \mathbf{f}_j$ for the paired latents against the projection of the residual stream onto $\mathbf{U}_{-1}$, observing a correlation of 0.99.

Finally, we show the $\mathbf{U}_{-1}$ dense signal reconstructed by this pair of antipodal latents is related to entropy regulation. This is motivated by previous work linking the bottom $\mathbf{W}_U$ subspace to a LayerNorm-based confidence regulation mechanism that is implemented in the model's final layer (Stolfo et al., 2024). First, we study the fraction of the residual stream norm along the last $\mathbf{W}_U$ singular vector $\mathbf{U}_{-1}$ (Figure 2b), which increases gradually towards later layers with a notable change at the final residual stream layer (just before unembedding). Then, we plot the variation of the residual stream projection along this direction against the model output entropy (Figure 2c), observing a significant correlation. These findings indicates that certain dense SAE latents are organized into nearly antipodal pairs that almost perfectly reconstruct a dense signal tied to entropy-related internal computation.

---

[2]Similarly to the computation for the residual stream principal components, the composition with the $\mathbf{W}_U$ singular vectors is calculated according to Eq. (1).

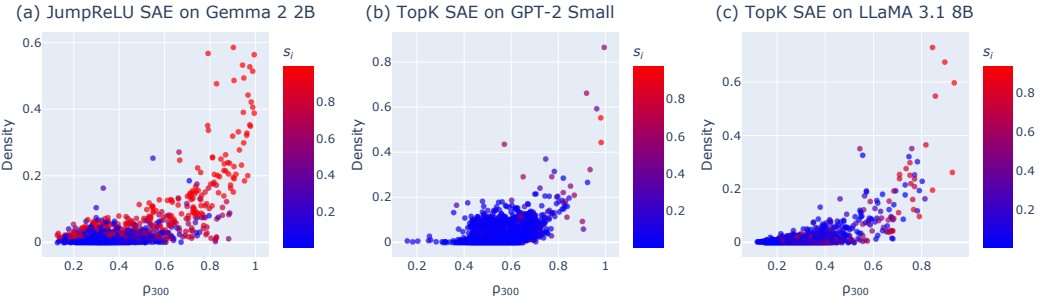

Figure 4: **Dense latents align with top residual components across models.** Latents with high activation density consistently exhibit strong alignment with the top 300 residual principal components ($\rho_{300}$) and high pairwise scores across SAEs.

### 3.3 ANTIPODAL PAIRING IN DENSE LATENTS

In this section, we investigate the connection between latent density and antipodal pairing. To quantify antipodal pairing, we introduce a metric based on cosine similarity. For each latent $i$, we compute the pairwise cosine similarities between its weights (both encoder and decoder) and those of all other latents. Then, we compute the maximum product of encoder and decoder cosine similarity across all pairs $(i, j)$ for all $i \neq j$. Specifically, we define a pairwise score $s_i$ for latent $i$ as

$$s_i := \max_{j \neq i} \left( \text{sim}\left(\mathbf{W}_{\text{enc}}^{(i)}, \mathbf{W}_{\text{enc}}^{(j)}\right) \cdot \text{sim}\left(\mathbf{W}_{\text{dec}}^{(i)}, \mathbf{W}_{\text{dec}}^{(j)}\right) \right),$$

where $\text{sim}(u, v)$ denotes the cosine similarity between vectors $u$ and $v$. This score reflects the extent to which latent $i$ forms an antipodal pairing with another latent: high values of $s_i$ indicate that there exists another latent $j$ with both encoder and decoder weights nearly opposite in direction to those of $i$.[3]

We then analyze the relationship between the pairwise score $s_i$ and the activation density of latent $i$. As shown in Figure 3, the majority of dense latents–particularly those with an activation frequency exceeding 0.3–exhibit pairwise scores greater than 0.9. This result highlights the observation that dense latents tend to organize into antipodal pairs, thereby facilitating the reconstruction of specific directions in the residual stream.

### 3.4 DIFFERENT MODELS AND SAES

We extend the analyses performed on the GemmaScope JumpReLU SAE to 32k-latent TopK SAEs (Gao et al., 2025) trained on the residual streams of GPT-2 Small and LLaMA 3.1 8B (He et al., 2024). Figure 4 illustrates the relationship between $\rho_{300}$, the pairwise score $s$, and the density for each latent of the last-layer SAEs of these models. The TopK SAEs exhibit an overall smaller number of latents with very high density (>0.2), most likely due to their larger number of latents (32k versus 16k

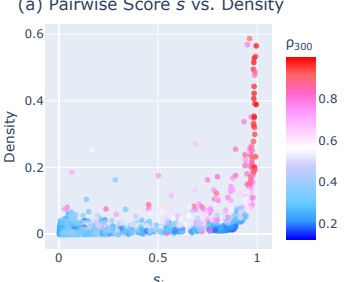

Figure 3: **Dense latents form antipodal pairs.** Latents with high activation density exhibit strong pairwise alignment, often aligning with top residual components ($\rho_{300}$).

in the Gemma Scope SAE). However, the trend remains consistent: latents with a high composition relative to the top $k = 300$ principal components of the residual stream also tend to exhibit high pairwise scores $s_i$ and high activation densities.

## 4 CONCLUSION

In this work, we demonstrate that dense latents in sparse autoencoders (SAEs) are purposefully allocated to reconstruct key directions in the residual stream. Our experiments on GemmaScope

---

[3]Although high values of $s$ could be produced by two nearly identical latents, retaining such a pair would be redundant–a scenario we do not observe. Evidence for this is provided in Appendix B, Figure 5c.

JumpReLU and TopK SAEs for GPT-2 Small and LLaMA 3.1 8B reveal that these latents align with the top residual stream principal components and the bottom singular vectors of the unembedding matrix $\mathbf{W}_U$–a subspace linked to internal signals such as entropy regulation–and are often organized into nearly antipodal pairs that jointly reconstruct specific directions. These findings suggest that dense latents capture mechanistic signals important to language model behavior, pointing to opportunities for refining SAE training objectives.

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

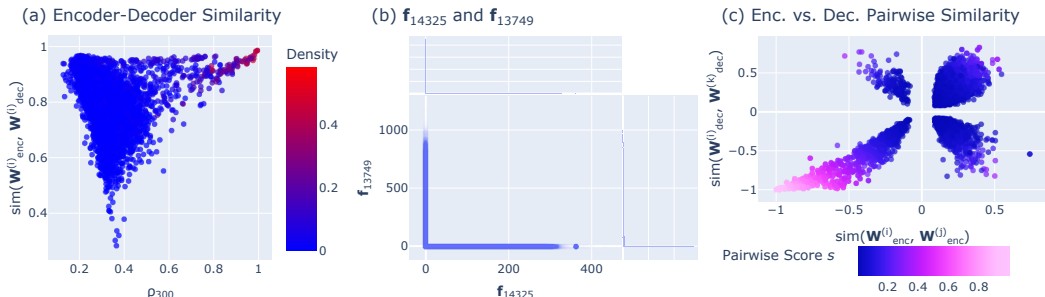

Figure 5: **Encoder-decoder similarity and antipodal pairs.** (a) High-density latents exhibit strong encoder-decoder alignment. (b) A dense latent pair rarely activates simultaneously. (c) High pairwise scores ($s$) occur when encoder and decoder weights are nearly opposite.

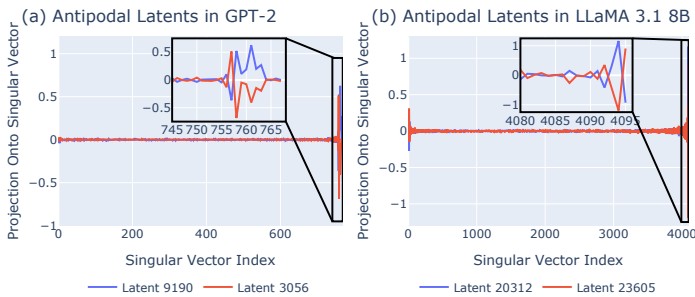

Figure 6: **Antipodal dense latents align with the bottom subspace of $\mathbf{W}_\text{U}$.** Projection of decoder weights for two dense latent pairs from TopK SAEs trained on GPT-2 Small (a) and LLaMA 3.1 8B (b), showing near-opposite alignment along the lowest singular vectors of the unembedding matrix.

## A    IMPLEMENTATION DETAILS

We carry out our experiments using the `TransformerLens` (Nanda & Bloom, 2022) and `SAELens` (Bloom et al., 2024) libraries. Our analyses use SAEs from Gemma Scope (Lieberum et al., 2024) and LLaMA Scope (He et al., 2024) for Gemma 2 2B and LLaMA 3.1 8B, respectively. For GPT-2 Small, we use the SAE family released by Gao et al. (2025).

## B    ADDITIONAL RESULTS

**Encoder-decoder Cosine Similarity.**    We analyze the similarity between the directions SAE latents read from and write onto. In Figure 5a, we plot the cosine similarity between each latent's encoder and decoder weights, $\text{sim}(\mathbf{W}_\text{enc}^{(i)}, \mathbf{W}_\text{dec}^{(i)})$. Interestingly, latents with high composition in the top residual principal components (which are also highly dense) exhibit strong encoder-decoder alignment, suggesting they both extract and reconstruct information along the same direction.

**Activation Patterns of Antipodal Pairs.** Figure 5b shows the activation values of the dense latent pair analyzed in §3.2 over 250k tokens. Despite their high activation frequency, they are virtually never active simultaneously.

**Pairwise Similarity Between Latents' Weights.**    In Figure 5c, we report for each latent $i$, the maximum-magnitude cosine similarity of its encoder and decoder weights with any other latent In particular, we show $\text{sim}(\mathbf{W}_\text{enc}^{(i)}, \mathbf{W}_\text{enc}^{(j)})$ and $\text{sim}(\mathbf{W}_\text{dec}^{(i)}, \mathbf{W}_\text{dec}^{(k)})$, where $j = \arg\max_{l \neq i}(|\text{sim}(\mathbf{W}_\text{enc}^{(i)}, \mathbf{W}_\text{enc}^{(l)})|)$ and $k = \arg\max_{l \neq i}(|\text{sim}(\mathbf{W}_\text{dec}^{(i)}, \mathbf{W}_\text{dec}^{(l)})|)$. We find that the pairwise score $s$ approaches 1 only when both encoder and decoder similarities are close to $-1$.

**Examples of Antipodal Latent Pairs.** We present two examples of antipodal dense latents from TopK SAEs trained on GPT-2 Small and LLaMA 3.1 8B. Figure 6 shows the projection of their decoder weights onto the respective model's unembedding matrix $\mathbf{W}_\text{U}$. These latents, with activation

densities of 0.86 and 0.09 for GPT-2 and 0.60 and 0.26 for LLaMA, exhibit strong composition with the bottom subspace of $\mathbf{W}_{\mathrm{U}}$.

