# OpenReview forum: "Antipodal Pairing and Mechanistic Signals in Dense SAE Latents"
_ICLR.cc/2025/Workshop/BuildingTrust — BuildingTrust_

### Official Review · Reviewer_XpvX · 2025-03-02
**Review of "Antipodal Pairing and Mechanistic Signals in Dense SAE Latents"**

**Rating:** 6
**Confidence:** 4

**Review:**

## **Summary**
This paper analyzes the dense latents of Sparse Autoencoders (SAEs). It studies the relationship between dense SAE latents and the language model's residual stream, revealing that SAE dense latents learn important signals to reconstruct directions in the residual stream. More precisely, it analyzes:  1) The alignment of dense latents with the top principal components of the residual stream.  2) The alignment of dense latents with the dark subspace of the unembedding matrix.  3) The connection between latent density and antipodal pairing.

## **Strengths & Weaknesses**

### **Strengths**
- This paper is well-written and presents a novel perspective in its analysis of dense latents in SAEs.
- The study examines dense latents from three different aspects, supported by sufficient experiments and demonstrations.
- The finding that dense latents are learned in a way that reconstructs key directions in the residual stream is particularly insightful.

### **Weaknesses**
- The main weakness of this study is that it does not provide an explanation for certain observed behaviors, which could be useful for exploring alternative solutions.
- Although the paper states that most latents are not activated frequently, Figure 4 (for GPT-2) shows that a larger number of latents have higher $\rho$ values, and a few latents have high pairwise scores and there is lack of explanation for that.
- Additionally, there are some minor typographical errors, such as:
  - In line 80, both terms are written as $W_{\text{enc}}$ instead of distinguishing between the encoder and decoder weights.
  - In line 198, a "." (period) is missing.

---

### Official Review · Reviewer_Wvcc · 2025-03-02
**Review of paper: Antipodal Pairing and Mechanistic Signals in Dense SAE Latents**

**Rating:** 6
**Confidence:** 3

**Review:**

The paper investigates the role of dense latents in Sparse Autoencoders (SAEs). The authors show that dense latents capture fundamental signals. They also show that most dense latents are arranged in antipodal pairs.

### Pros

- The paper provides a novel explanation for what information dense latents in SAEs contain, specifically for language models.
- There are detailed empirical results that support the authors' claims.

### Cons
- The paper is a bit hard to follow due to limited introduction on the background. It would be helpful to add some background information on dense latents, residual streams, etc.
- Lack of theoretical analyses. Consider adding more theoretical analyses to support the empirical results.
- Clarity in Mathematical Presentation – Some formulae and derivations, including Equation (1), could be explained more clearly for readers unfamiliar with SAE training dynamics.

---

### Official Review · Reviewer_4MVj · 2025-03-02
**Though this paper presents an interesting study of dense latents and their alignment with key model subspaces, it lacks a strong theoretical foundation and does not establish clear causal relationships.**

**Rating:** 5
**Confidence:** 4

**Review:**

The study explores the role of dense latents in SAEs, a topic relevant to interpretability research in LLMs.  The authors conduct detailed evaluations across different language models (Gemma 2, GPT-2, LLaMA 3.1). The use of principal component analysis (PCA), singular value decomposition (SVD), and SHAP analysis adds rigor to the study.

However, the authors claim that dense latents serve a "mechanistic role" in the model’s residual stream but provide little theoretical backing for why this phenomenon occurs. There is no discussion on whether these findings generalize across different training paradigms or model architectures. In addition, while the correlation between dense latents and residual stream variance is well-documented, correlation does not imply causation. The paper does not perform intervention-based experiments (e.g., ablating dense latents and observing performance degradation). It remains unclear if these latents **cause** important computations or merely reflect existing model dynamics.

Some experimental Designs should be clarified. For instance, the definition of "dense latents" is somewhat arbitrary (activating on 10%+ tokens). The threshold should be better justified. The synthetic setup lacks clarity: Are SAEs trained with different sparsity constraints to observe controlled effects? The role of antipodal latents is discussed qualitatively but lacks precise mathematical formulation.

Moreover, the paper does not compare SAEs with **alternative feature-extraction methods** (e.g., attention attribution, dictionary learning). Without such baselines, it is difficult to assess whether SAEs are uniquely capturing useful signals.

---

### Decision · Program_Chairs · 2025-03-02

Accept